# Improved Oral Health Status Is Associated with a Lower Risk of Venous Thromboembolism: A Nationwide Cohort Study

**DOI:** 10.3390/jpm13010020

**Published:** 2022-12-22

**Authors:** Jung-Hyun Park, Yoonkyung Chang, Jin-Woo Kim, Tae-Jin Song

**Affiliations:** 1Department of Oral and Maxillofacial Surgery, Mokdong Hospital, Ewha Womans University College of Medicine, Seoul 07985, Republic of Korea; 2Department of Neurology, Mokdong Hospital, Ewha Womans University College of Medicine, Seoul 07985, Republic of Korea; 3Department of Neurology, Seoul Hospital, Ewha Womans University College of Medicine, Seoul 07804, Republic of Korea

**Keywords:** periodontitis, oral hygiene, tooth brushing, venous thromboembolism, deep vein thrombosis, pulmonary thromboembolism

## Abstract

Oral health is reportedly associated with several systemic diseases, particularly cardiovascular diseases, through systemic inflammatory and thrombotic mechanisms. This study aimed to investigate the association between oral health status, oral hygiene behavior, and venous thromboembolism (VTE) in a nationwide, population-based cohort database in a longitudinal setting. Data of participants who underwent oral health screening by dentists between January and December 2003 (*n* = 2,415,963) were retrieved from the National Health Insurance Database of the Korean National Health Insurance Service. Periodontitis was identified using claims or oral health screening data. Periodontal pockets and the number of missing teeth were examined by dentists during oral health screenings. Data on oral hygiene behaviors (tooth brushing, dental visits, and dental scaling) were collected. VTE was defined as two or more claims of one of the following ICD-10 codes: deep (I80.2–80.3), pulmonary (I26, I26.0, I26.9), intra-abdominal (I81, I82, I82.2, I82.3), and other (I82.8, I82.9) VTE and concurrent medication (anticoagulants and antiplatelets). VTE was analyzed using the Cox proportional hazard model according to periodontitis, number of missing teeth, tooth brushing frequency, dental visits, and dental scaling. VTE occurred in 39,851 (1.8%) participants within a median of 17.0 (interquartile range 16.3–17.7) years. Periodontitis was associated with VTE (adjusted hazard ratio (HR), 1.2; 95% confidence interval (CI), 1.15–1.28; *p* < 0.001). An increased number of missing teeth was associated with an increased risk of VTE; the adjusted HR (versus participants without missing teeth) was 1.58 (95% CI, 1.46–1.71; *p* < 0.001, *p* for trend < 0.001) for participants with ≥15 missing teeth. Furthermore, tooth brushing ≥3 times a day was negatively correlated with VTE (adjusted HR, 0.67; 95% CI, 0.65–0.69; *p* < 0.001, *p* for trend < 0.001). Dental scaling within one year was associated with a significantly lower risk of VTE (adjusted HR, 0.95; 95% CI, 0.93–0.98; *p* < 0.001). Improved oral hygiene, including tooth brushing and dental scaling, may be associated with a decreased risk of VTE. Periodontitis and an increased number of missing teeth may increase the risk of VTE.

## 1. Introduction

Poor oral health conditions, such as periodontitis, dental caries, and tooth loss, are common health problems in the general population [1,2]. Microbial dysbiosis due to chronic oral disease, which begins with the accumulation of pathogenic microbial biofilms (plaques) around the gingiva, causes a chronic destructive inflammatory response surrounding periodontal tissue, resulting in systemic inflammation [3]. Poor oral health conditions adversely affect oral health and may be systematically associated with or trigger the occurrence of various diseases in the human body [4,5]. In addition, tooth loss and brushing have been reported to increase and decrease the risk of various systemic diseases such as diabetes, cardiovascular disease, certain cancers, and neurodegenerative diseases [6,7,8,9,10,11].

Venous thromboembolism (VTE) is a common condition with a high global disease burden that sometimes results in mortality [12]. The incidence of VTE is increasing owing to the global trend of transitioning to an aging society [13]. Cancer, antiphospholipid antibody syndrome, fractures, chronic inflammatory disorders, renal failure, and obesity have been suggested as risk factors for VTE [13]. However, information on preventable or correctable associations and risk factors for VTE is required [14].

Poor oral condition is associated with systemic inflammation, hypercoagulability, and platelet aggregation [15]. In addition, periodontitis is associated with the risk of VTE [16]. However, to date, few longitudinal studies have examined the association between general oral health and related behaviors, such as periodontitis, tooth loss, tooth brushing, and dental scaling, and VTE risk in the general population. The present study hypothesized that poor oral health status is positively correlated with VTE occurrence and that better oral health is associated with a lower risk of VTE. Therefore, this study aimed to longitudinally examine the association between VTE and oral health parameters in a population-based cohort.

## 2. Methods

### 2.1. Data Source

The National Health Insurance Database of the Korean National Health Insurance Service (NHIS), a public data source for the entire South Korean population, was used in this study [17]. The NHIS provides this research database to health researchers for access to useful data. The Korean government supervises and supports the NHIS, which is the sole insurance provider in Korea and covers nearly 97% of Koreans. The rest of the population is covered by the Medical Aid program, administered by the NHIS [1,2,18]. It is recommended that members of the NHIS undergo standardized health screening every 1–2 years. From the NHIS Database, adult participants (aged ≥ 20) who underwent oral health screenings between January 2003 and December 2003 (*n* = 2,415,963) were included in the present study (dataset number: NHIS-2022-01-313). The database includes the claims database for diagnosis, treatment, and prescription, as well as demographic and socioeconomic information. It also contains individual health screening information, including body mass index, blood pressure, laboratory tests, and questionnaires on lifestyle, such as oral hygiene behaviors. During health screenings, dentists examined the participants for dental problems, such as the number of missing teeth. The institutional review board approved this study (2021-07-034), and the requirement for informed consent was waived because of the anonymity of the data.

### 2.2. Study Population

Among the participants (*n* = 2,415,963), those with missing data for at least one variable of interest (*n* = 167,106) were excluded. Furthermore, participants with a history of VTE between January 2002 and the time of their oral health screening were excluded (*n* = 2187). Ultimately, 2,246,670 participants were included in this study (Figure 1).

### 2.3. Definition and Variables

The date of the oral health screening was set as the index date. Data on baseline characteristics, including age, sex, household income, and body mass index, were collected at the index date. Data on smoking habits, frequency of alcohol consumption per week, and frequency of regular physical exercise per week were obtained using questionnaires. Comorbidities were defined using claims of diagnosis (International Classification of Diseases, Tenth Revision, ICD-10), prescriptions, laboratory test results, or self-reported information in the questionnaire (Appendix B) [6,7,8,9,10,11,19].

The presence of periodontitis was defined according to the following criteria between January 2002 and the index date: (1) two or more claims of ICD-10 codes K052-054 (acute periodontitis (K052), chronic periodontitis (K053), and periodontitis (K054)) with at least one claim of related treatment codes (health claim codes: U0010, U1010, U1040, U1051-2, U1060, U2211, U2221-2, U2232-3, U2231,U2240, U4454-5, and U4660) or (2) detection of a periodontal pocket depth of 4 mm or more by a dentist during the oral health screening [3]. The number of missing teeth was assessed by a dentist during the oral health screening. The number of missing teeth, regardless of etiology, was classified as 0, 1–7, 8–14, or ≥15 [9,17]. Oral hygiene behaviors were collected as self-reported data during oral health screenings, including the frequency of tooth brushing per day, a dental visit for any reason within the past year, and dental scaling within the past year. When an oral health screening was performed more than twice, the most recent data were analyzed.

### 2.4. Study Outcomes

The main study outcome was VTE occurrence, which was defined based on a previous study as more than one claim of one of the following ICD-10 codes: deep vein thrombosis (I80.2–80.3), pulmonary VTE (I26, I26.0, I26.9), intra-abdominal VTE (I81, I82, I82.2, I82.3), and other VTE (I82.8, I82.9) and concurrent medication codes (anticoagulants and antiplatelet) [20]. The participants were followed-up from one day after the oral health screening until VTE or death occurred or until December 2020, whichever came first.

### 2.5. Statistical Analysis

The baseline characteristics of the periodontitis-positive and periodontitis-negative groups were compared using the chi-square test and independent t-test. Because the statistics for analyzing differences between groups are based on sample size, false positives are likely to occur with chi-square tests and independent t-tests for large-sized data sets. Thus, we used standardized differences, considering those >0.1 as noteworthy. Categorical and continuous variables are expressed as numbers (percentages) and mean ± standard deviation, respectively.

Kaplan–Meier survival curves and the log-rank test were used to evaluate the association of oral health status and oral hygiene behaviors with incident VTE risk. To estimate the incidence of VTE, the number of VTE cases was divided by the sum of person-years. To determine the effect of oral health parameters on VTE occurrence, the Cox proportional hazard regression was used, and hazard ratios (HRs) and 95% confidence intervals (CIs) were estimated. A multivariate regression model was constructed using adjustments for age, sex, body mass index, household income, alcohol consumption, smoking status, regular physical activity, and comorbidities (hypertension, diabetes mellitus, dyslipidemia, atrial fibrillation, cancer, renal disease, antiphospholipid antibody syndrome, and osteoporotic fractures). The oral health parameters were adjusted separately in the multivariate analysis because of multicollinearity. Subgroup analysis of the association between periodontitis and the occurrence of VTE was performed according to age, sex, and covariates. For the sensitivity analysis, further analyses were performed for each type of VTE (deep, pulmonary, intra-abdominal, and other VTE). Schoenfeld residuals were used to examine the assumptions of hazard proportionality. The proportional hazard assumption was not violated. All statistical analyses were performed using the Statistical Analysis System software (SAS version 9.2; SAS Institute, Cary, NC, USA). All values were considered statistically significant at *p* < 0.05.

## 3. Results

Among a total of 2,246,670 participants, the average age of the included participants was 42.3 ± 12.8 years, and 66.3% were male. A total of 14,100 (0.6%) participants had more than 15 missing teeth, 925,589 (41.2%) participants brushed their teeth more than 3 times a day, and 514,029 (22.9%) participants had dental scaling within the previous year. Table 1 shows the baseline characteristics and comparative analysis according to the presence of periodontitis.

VTE occurred in 39,851 (1.8%) participants within a median duration of 17.0 (interquartile range, 16.3–17.7) years. When each VTE was considered, deep vein thrombosis (15,175 (0.7%)), pulmonary (10,642 (0.5%)), intra-abdominal (18,338 (0.8%)), and other (16,829 (0.8%)) VTE occurred. Figure 2 shows the Kaplan–Meier survival curves of the participants free of VTE. The risk of incident VTE was higher in participants with periodontitis (*p* < 0.001), a higher number of missing teeth (*p* < 0.001), and a history of dental clinic visits within the previous year (*p* < 0.001). Moreover, better oral hygiene behaviors—a higher frequency of daily tooth brushing and a history of dental scaling—within the previous year were also associated with a decreased occurrence of VTE (*p* < 0.001) (Appendix A).

In the multivariate analysis, the presence of periodontitis was associated with the occurrence of VTE (adjusted HR, 1.21; 95% CI, 1.15–1.28; *p* < 0.001). An increase in the number of missing teeth was positively correlated with VTE occurrence; the adjusted HR (using participants with no missing teeth as reference) was 1.58 (95% CI, 1.46–1.71; *p* < 0.001, *p* for trend < 0.001) for participants with ≥15 missing teeth. Furthermore, a higher frequency of tooth brushing was negatively correlated with VTE occurrence. Compared to participants who brushed their teeth no more than once per day, those who brushed their teeth ≥ three times daily (adjusted HR, 0.67; 95% CI, 0.65–0.69; *p* < 0.001; *p* for trend < 0.001) had a decreased risk of VTE occurrence. Moreover, dental scaling within one year showed a significantly lower risk of VTE occurrence (adjusted HR, 0.95; 95% CI, 0.93–0.98; *p* < 0.001) (Table 2). In the subgroup analysis, the association between periodontitis and the occurrence of VTE was consistently noted regardless of the covariate (Appendix A), except for age.

In the sensitivity analysis, the association of the presence of periodontitis, number of missing teeth, frequency of tooth brushing, and dental scaling within one year with the occurrence of VTE was consistently noted in deep VTE, pulmonary VTE, intra-abdominal VTE, and other VTE. However, the association of dental scaling within one year with intra-abdominal VTE was not consistently noted (Appendix A).

## 4. Discussion

In the present study, both periodontitis and an increased number of missing teeth were oral health conditions associated with a higher risk of VTE occurrence. In contrast, better oral hygiene (dental scaling and frequent tooth brushing) was associated with a lower risk of VTE. These results are consistent with those of deep vein thrombosis and pulmonary, intra-abdominal, and other VTE.

Poor oral health status has systemic inflammatory consequences and is associated with various systemic diseases. Periodontitis reportedly increases the risk of cardiovascular and several vascular diseases [21,22]. Tooth loss, an indicator of poor oral health, is positively correlated with an increased risk of hypertension [23], stroke [24], and increased levels of thrombotic variables [25] which are associated with thrombus formation. According to a previous cross-sectional study, a higher prevalence of periodontitis was detected in patients with VTE than in healthy participants, suggesting a potential association between periodontitis and VTE [26]. The results of our study support the results of previous studies and are meaningful because they demonstrate the association between VTE and poor oral health in the general population in a longitudinal setting. Furthermore, another study showed that self-reported tooth loss due to periodontitis was associated with a higher risk of VTE. However, the association between periodontitis and VTE occurrence was not statistically significant [16]. The difference between these results and those of our study may be attributed to differences in the study design or population.

Conversely, behaviors that decrease oral inflammation, such as tooth brushing and professional dental care, are known to reduce the risk of certain systemic diseases. Frequent tooth brushing significantly reduces the occurrence of cardiovascular disease in patients with hypertension [10]. In addition, frequent tooth brushing attenuates the risk of diabetes [2], and dental scaling reduces the risk of end-stage renal disease [5]. A recent meta-analysis reported that dental scaling reduces the risk of atrial fibrillation [27]. Our study is consistent with these previous studies and provides new information on the association between VTE and oral hygiene behaviors, as we have shown that people with better oral hygiene behaviors have a reduced risk of VTE.

Although a direct causal relationship between oral health status and VTE occurrence could not be determined in the present study, the following hypothesis may explain this association: (1) Periodontal pathogens directly invade the vasculature and induce thrombus formation. Periodontitis is frequently associated with bacteremia, and periodontal bacterial DNA has been detected in coronary arterial plaques and intraluminal thrombi [28]. Oral bacteria have a platelet-aggregating ability and may directly cause thrombus formation [29,30]. (2) Inflammation induced by periodontitis produces pro-inflammatory cytokines that trigger thrombus formation [31]. A possible association has been demonstrated between VTE and several pro-inflammatory cytokines, such as interleukin-6, interleukin-8, and C-reactive protein (CRP) [32,33,34,35,36]. These cytokines have been reported to play an important role in thrombus formation by inducing the expression of tissue factors that promote a procoagulant state [31,37]. CRP and fibrinogen levels are reportedly higher in patients with chronic and aggressive periodontitis than in healthy participants [38], and tooth loss is associated with increased fibrinogen and factor VIII levels, supporting this hypothesis [25]. Conversely, improved oral hygiene through brushing and dental scaling appears to reduce the risk of VTE through changes in the oral microbiome and reduced inflammatory burden. Since periodontal pathogens can be significantly reduced by dental scaling or brushing [39], better oral hygiene may reduce thrombus formation caused by direct bacteremia. A reduced inflammatory response through improved oral hygiene may reduce the risk of VTE by reducing the production of pro-inflammatory cytokines. Frequent tooth brushing is reportedly associated with decreased concentrations of both CRP and fibrinogen [40]. A previous meta-analysis found that, in patients with periodontitis, periodontal treatment lowered the levels of systemic inflammatory markers, which also supports the reduction in inflammatory burden and VTE risk [41].

The present study has several limitations. First, there may have been residual confounders or inflammatory biomarkers, such as D-dimer and high-sensitivity CRP, which affected VTE occurrence and were not included in our data set. Second, the participants in the present study were Korean, and the results may differ for other populations. Third, because information on periodontal loss of attachment was lacking in the NHIS data, periodontitis severity was not considered. Fourth, there was no detailed information regarding the cause of tooth loss in the database; therefore, tooth loss due to periodontitis could not be assessed separately. Fifth, because oral health behaviors were based on a self-reported questionnaire, there may have been response bias, such as social desirability bias. Sixth, our study could not suggest a causal relationship as it was a retrospective observational study. Finally, because the mechanism of cerebral VTE differs from that of general VTE, it was not included in the data set from the beginning. However, the strength of the present study is that we used a long-tracked, large, nationally representative database to elucidate the link between oral health parameters and VTE occurrence.

The results of this study provide unique evidence supporting the importance of maintaining better oral health in VTE prevention. These findings may be considered when planning public health strategies for VTE prevention. Further studies are needed to confirm the association between chronic oral inflammation and VTE.

## 5. Conclusions

Oral health management, including frequent tooth brushing and professional dental scaling, appeared to be associated with a decreased risk of VTE, including deep, pulmonary, intra-abdominal, and other VTE. Poor oral health, such as periodontitis and an increase in the number of missing teeth, may increase the incidence of VTE.

## Figures and Tables

**Figure 1 jpm-13-00020-f001:**
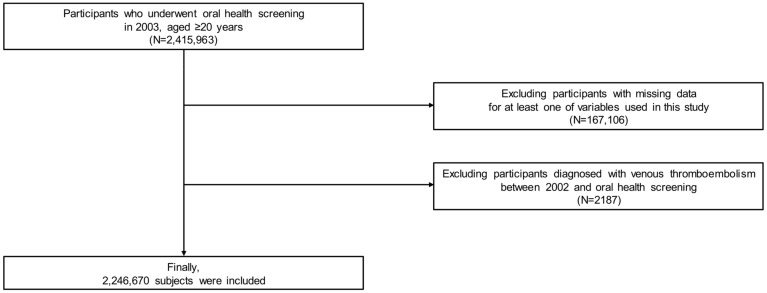
Flow chart of study participants.

**Figure 2 jpm-13-00020-f002:**
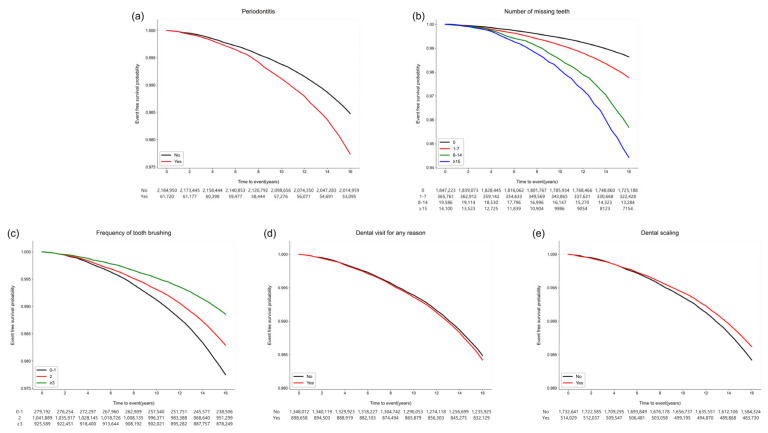
Kaplan–Meier survival curves for the occurrence of VTE according to oral health status and oral hygiene behaviors. (**a**) Periodontitis (*p* < 0.001). (**b**) Number of missing teeth (*p* < 0.001). (**c**) Frequency of tooth brushing (times/per day) (*p* < 0.001). (**d**) Dental visit for any reason within the previous year (*p* < 0.001). (**e**) Dental scaling within the previous year (*p* < 0.001).

**Table 1 jpm-13-00020-t001:** Baseline characteristics of subjects according to the presence of periodontitis.

Variable	Total	Periodontitis (-)	Periodontitis (+)	*p*-Value	Standardized Difference
Number of participants (%)	2,246,670	2,184,950 (97.3)	61,720 (2.7)		
Age, years	42.33 ± 12.79	42.16 ± 12.74	48.22 ± 13.12	<0.001	0.47
Sex				<0.001	0.09
Male	1,489,116 (66.3)	1,445,744 (66.2)	43,372 (70.3)		
Female	757,554 (33.7)	739,206 (33.8)	18,348 (29.7)		
Body mass index (kg/m^2^)	23.55 ± 14.41	23.55 ± 14.59	23.8 ± 4.27	<0.001	0.02
Household income				<0.001	−0.05
Q1, lowest	588,309 (26.2)	570,419 (26.1)	17,890 (29.0)		
Q2	813,814 (36.2)	792,146 (36.3)	21,668 (35.1)		
Q3	587,710 (26.2)	572,296 (26.2)	15,414 (25.0)		
Q4, highest	256,837 (11.4)	250,089 (11.5)	6748 (10.9)		
Smoking status				<0.001	0.08
Never	1,259,767 (56.1)	1,227,498 (56.2)	32,269 (52.3)		
Former	241,102 (10.7)	234,028 (10.7)	7074 (11.5)		
Current	745,801 (33.2)	723,424 (33.1)	22,377 (36.3)		
Alcohol consumption (days/week)				<0.001	0.13
None	1,503,378 (66.9)	1,464,803 (67.0)	38,575 (62.5)		
1–4	684,835 (30.5)	665,055 (30.4)	19,780 (32.1)		
≥5	58,457 (2.6)	55,092 (2.5)	3365 (5.5)		
Regular physical activity (days/week)				<0.001	−0.03
None	1,170,666 (52.1)	1,136,694 (52.0)	33,972 (55.0)		
1–4	914,044 (40.7)	891,416 (40.8)	22,628 (36.7)		
≥5	161,960 (7.2)	156,840 (7.2)	5120 (8.3)		
Comorbidities					
Hypertension	419,789 (18.7)	403,922 (18.5)	15,867 (25.7)	<0.001	0.17
Diabetes mellitus	170,043 (7.6)	162,496 (7.4)	7547 (12.2)	<0.001	0.16
Dyslipidemia	283,620 (12.6)	274,749 (12.6)	8871 (14.4)	<0.001	0.05
Atrial fibrillation	3682 (0.2)	3529 (0.2)	153 (0.3)	<0.001	0.02
Cancer	20,651 (0.9)	19,878 (0.9)	773 (1.3)	<0.001	0.03
Renal disease	11,258 (0.5)	10,826 (0.5)	432 (0.7)	<0.001	0.03
Antiphospholipid syndrome	2786 (0.1)	2658 (0.1)	128 (0.2)	0.001	0.01
Osteoporotic fracture	12,556 (0.6)	12,042 (0.6)	514 (0.8)	<0.001	0.03
Oral health status					
Number of missing teeth				<0.001	0.35
0	1,847,223 (82.2)	1,805,600 (82.6)	41,623 (67.4)		
1–7	365,761 (16.3)	348,314 (15.9)	17,447 (28.3)		
8–14	19,586 (0.9)	17,639 (0.8)	1947 (3.2)		
≥15	14,100 (0.6)	13,397 (0.6)	703 (1.1)		
Oral hygiene behaviors					
Frequency of tooth brushing (times/day)				<0.001	−0.20
0–1	279,192 (12.4)	268,584 (12.3)	10,608 (17.2)		
2	1,041,889 (46.4)	1,010,843 (46.3)	31,046 (50.3)		
≥3	925,589 (41.2)	905,523 (41.4)	20,066 (32.5)		
Dental visit for any reason				<0.001	−0.02
No	1,348,012 (60.0)	1,310,281 (60.0)	37,731 (61.1)		
Yes	898,658 (40.0)	874,669 (40.0)	23,989 (38.9)		
Dental scaling				<0.001	−0.11
No	1,732,641 (77.1)	1,682,402 (77.0)	50,239 (81.4)		
Yes	514,029 (22.9)	502,548 (23.0)	11,481 (18.6)		

Q, Quartile. *p*-values were determined using the chi-squared test. Data are expressed as mean ± standard deviation or n (%).

**Table 2 jpm-13-00020-t002:** The risk of venous thromboembolism according to oral health status and oral hygiene behaviors.

	Number of Participants	Number of VTE Events	VTE Rate (%) (95% CI)	Person-Years	Incidence Rate (Per 1000 Person-Years)	Adjusted HR (95% CI)	*p*-Value
Oral health status							
Periodontitis							
No	2,184,950	38,247	1.75 (1.73, 1.77)	35,662,664.64	1.07	1 (reference)	
Yes	61,720	1604	2.60 (2.47, 2.73)	986,978.23	1.63	1.21 (1.15, 1.28)	<0.001
Number of missing teeth							
0	1,847,223	29,045	1.57 (1.55, 1.59)	30,312,075.99	0.96	1 (reference)	
1–7	365,761	9260	2.53 (2.48, 2.58)	5,870,772.11	1.58	1.38 (1.35, 1.42)	<0.001
8–14	19,586	860	4.39 (4.10, 4.68)	284,308.82	3.03	1.59 (1.49, 1.71)	<0.001
≥15	14,100	686	4.87 (4.50, 5.23)	182,485.93	3.76	1.58 (1.46, 1.71)	<0.001
Oral hygiene behaviors							
Frequency of tooth brushing (times/day)							
0–1	279,192	7040	2.52 (2.46, 2.58)	4,418,588.01	1.59	1 (reference)	
2	1,041,889	20,645	1.98 (1.95, 2.01)	16,982,107.23	1.22	0.87 (0.84, 0.89)	<0.001
≥3	925,589	12,166	1.31 (1.29, 1.34)	15,248,947.62	0.80	0.67 (0.65, 0.69)	<0.001
Dental visit for any reason							
No	1,348,012	23,505	1.74 (1.72, 1.77)	21,935,911.19	1.07	1 (reference)	
Yes	898,658	16,346	1.82 (1.79, 1.85)	14,713,731.68	1.11	1.05 (1.03, 1.07)	<0.001
Dental scaling							
No	1,732,641	31,685	1.83 (1.81, 1.85)	28,190,468.22	1.12	1 (reference)	
Yes	514,029	8166	1.59 (1.55, 1.62)	8,459,174.65	0.97	0.95 (0.93, 0.98)	<0.001

The multivariable model included sex, age, body mass index, income level, smoking, alcohol consumption, regular physical activity, hypertension, diabetes mellitus, dyslipidemia, atrial fibrillation, cancer, renal disease, antiphospholipid syndrome, and osteoporotic fracture. CI, confidence interval; HR, hazard ratio.

## Data Availability

The data used in this study are available in the National Health Insurance Service (NHIS) database; however, restrictions apply to the public availability of these data used under license for the current study. Requests for access to NHIS data can be made through the National Health Insurance Sharing Service homepage (http://nhiss.nhis.or.kr/bd/ab/bdaba021eng.do, accessed on 22/11/2022). To access the database, a completed application form, research proposal, and application for approval from the institutional review board should be submitted to the inquiry committee of research support in the NHIS for review.

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
