# Peer review of "Improved Oral Health Status Is Associated with a Lower Risk of Venous Thromboembolism: A Nationwide Cohort Study"

_jpm, 2022, doi:10.3390/jpm13010020_

Round 1
Reviewer 1 Report
I am applying for approval of the manuscript in its entirety - this is one of the best studies I have been able to review. The sheer number of population data used in this retrospective study and high-value statistical analysis indicates the publication's highest value.
From the reviewer's office standpoint, I would like to ask you to add information about the affiliation of the first author (missing reference after name), I would also ask you to include after the section the limitations of the study, which in the authors' opinion is advisable in further scientific activity (global or national) in the perspective of the results obtained.
Author Response
Comment 1. I am applying for approval of the manuscript in its entirety - this is one of the best studies I have been able to review. The sheer number of population data used in this retrospective study and high-value statistical analysis indicates the publication's highest value.
From the reviewer's office standpoint, I would like to ask you to add information about the affiliation of the first author (missing reference after name),
⇒ Author response 1: We appreciate the time and effort that you have dedicated to providing feedback on our manuscript. Thank you for highlighting this missing information. The affiliation of the first author has been added to the revised manuscript.
Comment 2. I would also ask you to include after the section the limitations of the study, which in the authors' opinion is advisable in further scientific activity (global or national) in the perspective of the results obtained.
⇒ Author response 2: Thank you for this suggestion. We have included a paragraph that details our recommendations for future research based on our study's findings.
“The results of this study provide unique evidence supporting the importance of maintaining better oral health in VTE prevention. These findings may be considered when planning public health strategies for VTE prevention. Further studies are needed to confirm the association between chronic oral inflammation and VTE.”
Please see these changes on page 8, lines 263-266.
Reviewer 2 Report
Comments and Suggestions for Authors. The authors present a nationwide cohort study on relation between oral health and the risk of occurrence of thromboembolism. The main aim is very relevant for journal personalized medicine, but should be improved regrading methodology, data collection association, dentistry criteria used, nomenclature (oral health status or oral behavior criteria needs to be clear) and methodology applied to dentistry data.
If dentistry data is not clear, how can we get association or not with VTE, or other inflammatory and systemic diseases?
Oral health screening were collected in 2003, that is, about 20 years ago.
Title should needs to be improved and clear, according the conclusions sentence in line 40, and lines 109 to 119, and figure 2 (line 173) and table 2 (lines 190, 191). Oral health status instead of oral hygiene. Suggestion to authors “Improved oral health status and the risk of venous ….”
Needs to uniform the nomenclature in abstract and main text, according the Introduction description between lines 47 -55 and line 63, and methodology applied (lines 109 to 119). Data criteria for measuring oral health behavior (were self-reported- line 119) are completely distinct of data criteria for measuring, by evaluation, periodontite (oral health pathology), missing tooth aetiology or even dental scaling.
Abstract is some confusing regarding the dentistry data applied, methodology applied and oral diagnosis, treatments and self-reported terminology . Poor Oral hygiene behavior is related with periodontitis, but “tooth brushing” is the only one criteria described (and not confirmed by any oral hygiene index, even the Greene and Vermion index) for measuring, a self-report data about oral hygiene and oral hygiene behavior.
Line 18 – Please remove the symbol and the sentence. Add the sentence if necessary in the end of the manuscript in chapter “authors contribution”.
Line 106 and 107- Please add the cited 7 references of ICD, of appendix 1, in order to include all the references in the same chapter.
Line 109 and 110 – Needs to be clear and to re-write the sentence: “The presence of periodontitis…..” Please clarify and re-write the sentence regarding the evaluation assessment- What periodontal index examination was performed, during examination?
Line 113 and 114 – Please clarify and re-write the sentence regarding the evaluation assessment: “Dental caries and number of missing teeth” – What dental index was used? The WHO criteria? DMF teeth? Or another one?
Table 2 – Suggestion to add in the table line: “ VTE” – Number of VTE events and to “VTE rate (%)”
Discussion chapter needs to be improved regarding some sentences and statements that are not demonstrated, although results suggest correlation .
Line 201- Suggestion to improve the line 201 and 202- presence of periodontitis and increased number of missing teeth are oral health conditions associated with higher risk of VTE occurrence.
Line 211- Periodontitis was more prevalent….We-write more carefully; It should be clear the index to measure this prevalence.
Line 215 and 216 – In the present study, and according to line 115, the number of missing was registered regardless of its etiology, was classified as 0, 1–7, 8–14, or ≥15. This must be stared. Tooth lost is multifactorial an can be caused by severam dental decay conditions, such us, carious process, tooth wear, not only by periodontal aetiology; The present study did not assess the missed teeth caused by periodontal disease .
An accurate and editing of English language and style should be performed.
References: The cited references are mostly (about 46%) recent publications (within the last 5 years) and relevant. However, the reference list must follow the ACS style guide and Reference Formatting Guide, namely the references of journals and standards. Please review and correct the formatting style, The DOI should be added in some references list.
Author Response
Comment 1. The authors present a nationwide cohort study on relation between oral health and the risk of occurrence of thromboembolism. The main aim is very relevant for journal personalized medicine, but should be improved regrading methodology, data collection association, dentistry criteria used, nomenclature (oral health status or oral behavior criteria needs to be clear) and methodology applied to dentistry data. If dentistry data is not clear, how can we get association or not with VTE, or other inflammatory and systemic diseases? Oral health screening were collected in 2003, that is, about 20 years ago.
⇒ Author response 1: I appreciate the time and effort you have dedicated to providing feedback on our manuscript and am grateful for the insightful comments on our paper. This longitudinal study aimed to investigate the association between oral health status, oral hygiene behaviors, and venous thromboembolism (VTE). To determine this association, we used oral health screening data collected 20 years ago. We have addressed your valid concerns regarding the methodology and terminology in the following comments.
Comment 2. Title should needs to be improved and clear, according the conclusions sentence in line 40, and lines 109 to 119, and figure 2 (line 173) and table 2 (lines 190, 191). Oral health status instead of oral hygiene. Suggestion to authors “Improved oral health status and the risk of venous ….”
⇒ Author response 2: Thank you for your suggestion. We have changed “oral hygiene” to “oral health status” in our title.
Comment 3. Needs to uniform the nomenclature in abstract and main text, according to the Introduction description between lines 47 -55 and line 63, and methodology applied (lines 109 to 119). Data criteria for measuring oral health behavior (were self-reported- line 119) are completely distinct of data criteria for measuring, by evaluation, periodontitis (oral health pathology), missing tooth aetiology or even dental scaling.
⇒ Author response 3: As you pointed out, oral hygiene behavior is a different variable to oral health status. We have made changes throughout our manuscript to ensure consistent use of the terms “oral health status” and “hygiene behavior.” Please see our changes on:
Line 18: from “oral health estimates and behavior” to “oral health status and oral hygiene behavior”
Line 60: from “Poor oral hygiene and periodontitis” to “Poor oral condition”
Line 65: from “poor oral hygiene” to “poor oral health status”
Line 202: from “Periodontitis and poor oral health” to “Poor oral health status”
Line 223: from “oral health behaviors” to “oral hygiene behaviors”
Line 262: from “oral health” to “oral health parameters”
Comment 4. Abstract is some confusing regarding the dentistry data applied, methodology applied and oral diagnosis, treatments and self-reported terminology. Poor Oral hygiene behavior is related with periodontitis, but “tooth brushing” is the only one criterion described (and not confirmed by any oral hygiene index, even the Greene and Vermion index) for measuring, a self-report data about oral hygiene and oral hygiene behavior.
⇒ Author response 4: Thank you, we agree that our terminology could be clearer. As such, we have made changes to our terminology, which we believe enhances the clarity of the abstract.
Additionally, we have included further information which should clarify the oral health parameters measured.
“Periodontitis was identified using claims or oral health screening data. Periodontal pockets and number of missing teeth were examined by dentists during oral health screening. Data on oral hygiene behaviors (tooth brushing, dental visits, and dental scaling) were collected.”
Please see these changes made on page 1, lines 16-39.
Comment 5. Line 18 – Please remove the symbol and the sentence. Add the sentence if necessary in the end of the manuscript in chapter “authors contribution”.
⇒ Author response 5: Thank you for your suggestion. We have removed the symbol and sentence from line 18 and placed it under ‘author contributions.’
Comment 6. Line 106 and 107- Please add the cited 7 references of ICD, of appendix 1, in order to include all the references in the same chapter.
⇒ Author response 6: Thank you for this suggestion. The references in Appendix 1 have been added to the main manuscript.
Please see these changes on page 3, line 102.
Comment 7. Line 109 and 110 – Needs to be clear and to re-write the sentence: “The presence of periodontitis…..” Please clarify and re-write the sentence regarding the evaluation assessment- What periodontal index examination was performed, during examination?
⇒ Author response 7: Thank you for raising this issue. One of the criteria used to determine the presence of periodontitis was claim data from insurance services (ICD-10 codes K052-054). The other criterion was the oral health screening information collected during the periodontal pocket examination by a dentist. Assuming that there are no pockets deeper than 3 mm in the healthy periodontium, at least one pocket with a probing depth of 4 mm or more was defined as a periodontitis case. This information has been added to the Methods section, along with references.
“The presence of periodontitis was defined according to the following criteria between January 2002 and the index date: 1) two or more claims of ICD-10 codes K052-054 (acute periodontitis [K052], chronic periodontitis [K053], and periodontitis [K054]) with at least one claim of related treatment codes (health claim codes: U0010, U1010, U1040, U1051-2, U1060, U2211, U2221-2, U2232-3, U2231,U2240, U4454-5, and U4660), or 2) detection of a periodontal pocket depth of 4 mm or more by a dentist during oral health screening.”
Please see these changes on page 3, lines 103-108.
Comment 8. Line 113 and 114 – Please clarify and re-write the sentence regarding the evaluation assessment: “Dental caries and number of missing teeth” – What dental index was used? The WHO criteria? DMF teeth? Or another one?
⇒ Author response 8: Thank you for raising this issue; we are happy to clarify this information. The number of dental caries requiring treatment during oral health screening was counted by dentists without using a specific caries index. Dental caries may be associated with systemic diseases; however, the number of dental caries was not included in the analysis because the relationship between dental caries and systemic inflammation remains unclear. Therefore, the description of dental caries was unnecessary in this case, and we have deleted this information.
Comment 9. Table 2 – Suggestion to add in the table line: “ VTE” – Number of VTE events and to “VTE rate (%)”
⇒ Author response 9: Thank you for your recommendation. These changes have been made according to your suggestion.
Comment 10. Discussion chapter needs to be improved regarding some sentences and statements that are not demonstrated, although results suggest correlation.
Line 201- Suggestion to improve the line 201 and 202- presence of periodontitis and increased number of missing teeth are oral health conditions associated with higher risk of VTE occurrence.
⇒ Author response 10: Thank you for your suggestions. We have made changes to this sentence to clearly outline the correlation between oral health status and VTE occurrence.
“In the present study, both periodontitis and an increased number of missing teeth were oral health conditions associated with a higher risk of VTE occurrence.”
Please see our changes on page 7, lines 197-198.
Comment 11. Line 211- Periodontitis was more prevalent….We-write more carefully; It should be clear the index to measure this prevalence.
⇒ Author response 11: Thank you for highlighting this issue. This sentence was based on the results of a previous cross-sectional study. We have edited the sentence accordingly for clarity.
“According to a previous cross-sectional study, a higher prevalence of periodontitis was detected in patients with VTE than in healthy participants, suggesting a potential association between periodontitis and VTE.”
Please see our changes on page 7, lines 207-209.
Comment 12. Line 215 and 216 – In the present study, and according to line 115, the number of missing was registered regardless of its etiology, was classified as 0, 1–7, 8–14, or ≥15. This must be stared. Tooth lost is multifactorial an can be caused by several dental decay conditions, such us, carious process, tooth wear, not only by periodontal aetiology; The present study did not assess the missed teeth caused by periodontal disease.
⇒ Author response 12: Thank you for raising this issue. We agree that tooth loss due to periodontitis needs to be considered separately, as tooth loss can have various causes. However, as the database used in this study had information only on claims data and oral health screening, it was not possible to determine the cause of tooth loss. We have added a sentence explaining this study limitation to the Discussion section.
“Fourth, there was no detailed information regarding the cause of tooth loss in the database; therefore, tooth loss due to periodontitis could not be assessed separately.”
Please see our changes on page 8, lines 254-256.
Comment 13. An accurate and editing of English language and style should be performed.
⇒ Author response 13: Thank you for this suggestion. We have had our manuscript carefully reviewed by an English-language editing service.
Comment 14. References: The cited references are mostly (about 46%) recent publications (within the last 5 years) and relevant. However, the reference list must follow the ACS style guide and Reference Formatting Guide, namely the references of journals and standards. Please review and correct the formatting style, The DOI should be added in some references list..
⇒ Author response 14: Thank you for your suggestions regarding our references. These have been carefully reviewed and corrected according to journal style.